# Quality Characteristics and Antioxidant Activities of Six Types of Korean White Wine

**DOI:** 10.3390/foods12173246

**Published:** 2023-08-29

**Authors:** Sae-Byuk Lee

**Affiliations:** 1School of Food Science and Biotechnology, Kyungpook National University, 80 Daehakro, Daegu 41566, Republic of Korea; lsbyuck@knu.ac.kr; Tel.: +82-53-950-7749; 2Institute of Fermentation Biotechnology, Kyungpook National University, 80 Daehakro, Daegu 41566, Republic of Korea

**Keywords:** white wine, Cheongsoo grape, antioxidant activity, white grape, wine

## Abstract

The cultivation of European grape cultivars suitable for winemaking in Korea presents challenges due to factors such as climate, soil conditions, precipitation, and sunlight. Consequently, Korea has traditionally resorted to adding sugar to its wine production to counteract the low sugar content in Korean grapes, yielding lower-quality wines. However, recent success in the cultivation of five European grape cultivars and the development of the domestic grape cultivar Cheongsoo have increased the possibility of achieving high-quality Korean wines. This study aimed to explore the potential of European grape cultivars and Cheongsoo as wine grapes in Korea. This study also conducted sensory evaluation and analyzed the physicochemical properties of the grapes and wines, including antioxidant capacity and color. Despite originating from the same vineyard, the composition of grapes and wines, including volatile aromatic compounds, significantly differed among the grape cultivars. In particular, Vidal wine exhibited superior antioxidant capacity compared with other wines. Moreover, Cheongsoo wine showed higher levels of essential volatile aromatic compounds, such as monoterpenes, than other wines. Sensory evaluation of these two wines also revealed excellent results. In conclusion, these findings hold promise for enhancing the diversity of Korean white wine and fostering growth in the wine industry.

## 1. Introduction

Wine, a globally consumed alcoholic beverage, has been considered a favorite drink for a long time and is largely classified into red, white, and rosé wines based on its color. To prepare red wine, fully ripe red grapes are harvested and immediately crushed to extract the skin, flesh, and seeds. This mixture then undergoes fermentation and maceration [1]. In contrast, to prepare white wine, slightly unripe white grapes are harvested. The skin is removed in advance, and the grapes are gently pressed to prevent seed breakage. The obtained fruit juice is finally fermented at a low temperature (10–15 °C), resulting in relatively lower levels of tannin than those in red wine. This process impacts a refreshing taste and a bright golden color [2]. Rosé wines, which exhibit a delicate pink shade, are prepared by rapidly removing grape skins at the beginning of the fermentation process. They are composed of a mixture of both “black” and “white” grape cultivars, employing techniques similar to those in white wine production [2]. Wine contains various components, including sugars, acids, tannins, minerals, organic acids, and volatile aromatic compounds. Notably, it contains a diverse range of phenolic compounds, such as phenolic acid, resveratrol, flavonol, flavan-3-ol, and anthocyanin. These compounds are well-known for their functional properties, such as anti-inflammatory, antiobesity, antioxidant, antiaging, cardioprotective, and antiallergic effects [3,4,5,6,7]. Although white wine appears to possess fewer health benefits due to its relatively lower polyphenol content than that in red wine, several researchers who conducted extensive epidemiological surveys on human health revealed no significant difference between red and white wines in terms of preventing coronary diseases such as type 2 diabetes [8,9,10,11,12].

Wine quality is distinctly influenced by several factors such as grape cultivar, cultivation technology, cultivation region, climate, aging conditions, handling techniques, and microbes [13,14,15,16,17]. Among these, the grape cultivar stands out as the foundational element in winemaking, exerting the most substantial impact on the final wine quality, composition, and aromatic compounds like volatile acids, esters, higher alcohols, terpenes, and more [7,14,18]. However, wine grape cultivation in Korea faces limitations. Various wine grape cultivars such as Cabernet Sauvignon, Merlot, and Sauvignon Blanc, which are commonly grown in Europe and the U.S., struggle to thrive in Korea’s hot, humid, and rainy summer climate. This unfavorable environment leads to issues like inadequate flower-bud differentiation, poor coloration, and fungal diseases [19,20,21]. Hence, most Korean winemakers have utilized various table grapes such as Campbell Early, Muscat Bailey A, and the wild grape Meoru. These cultivars are well-suited to Korea’s climate and soil conditions, enabling the production of red wine. In contrast, the cultivation of white grapes, such as Geobong, Shine Musket, and Alexandria is increasing in Korea; however, these grapes are not commonly used for wine production. In 1993, the National Institute of Horticultural and Herbal Science in Korea bred the grape cultivar ‘Cheongsoo’ through a plant cross between the grape cultivars Seibel 9110 and Himrod Seedless; this cultivar is widely used for table and wine grapes because of its distinct golden color and attractive fruit flavor [22,23]. Nonetheless, it is important to explore other grape cultivars for producing white wine to meet the increasing demand among Koreans. Although several researchers have recently confirmed the potential of volatile aromatic compounds in 11 red wines and 2 white wines from different international grapes grown in Korea as wine grape cultivars [19,24], additional research is warranted on the development of various white grape cultivars. Based on the recent successful attempts to cultivate five foreign white grape cultivars, including Chardonnay, Sémillon, Seibel, Kerner, and Vidal, after several years of effort in Yeongcheon, Gyeongbuk Province, they may be explored as grape cultivars for winemaking.

In this study, the physicochemical, functional, and organoleptic properties of six white wines using the ‘Cheongsoo’ cultivar and five international white grape cultivars grown in Korea were investigated. This study aims to expand the diversity of Korean white wine and the domestic white wine market.

## 2. Materials and Methods

### 2.1. Strain, Material, and Culture Conditions

*Saccharomyces cerevisiae* EC-1118, an industrial wine yeast strain, was utilized for the wine fermentation process. Yeast cells were cultivated at 30 °C under shaking conditions (150 rpm) in YPD media comprising 1% yeast extract, 2% bacto-peptone, and 2% glucose. The cells were harvested for wine fermentation. This study examined the following six white grape cultivars: Chardonnay Blanc (CB), Sémillon (SM), Seibel 9110 (SB), Kerner (KN), Vidal Blanc (VB), and Cheongsoo (CS) (Figure 1). These grape cultivars were cultivated using “SO4 (Selection Oppenheim 4)” rootstock in the same vineyard located in Yeongcheon, Gyeongbuk Province (35.9733 °N latitude and 128.9385 °E longitude). The training system was double cordon, and the vine spacing was 2.0 m between rows and 2.0 m within each row. Phytosanitary treatments were carried out based on the cultivation season for grape cultivation, and the irrigation treatment was conducted during the grape growth period (March to August). They were harvested simultaneously during mid-September. Subsequently, ripe and healthy grapes were meticulously handpicked for the lab-scale winemaking process.

### 2.2. Wine Fermentation

The white grapes were washed, stemmed, and pressed using Waterpress (Hydro 80, Zambelli Enotech, Camisano Vicentino, Italy) at a pressure level of 300 kPa to obtain grape must for the wine fermentation process. Before the inoculation of wine yeast, 200 mg/L of potassium metabisulfite (K_2_S_2_O_5_) was added to 5 L of grape must to prevent bacterial contamination. The yeast cells, which were cultured overnight in YPD broth at 30 °C under shaking conditions (150 rpm), were inoculated into 1 L flasks containing 250 mL of grape must. This mixture was incubated at 30 °C under shaking conditions (150 rpm) for 2 days until the yeast cell count reached ~10^8^ CFU/mL. Next, the yeast cells were inoculated into a 20 L fermentation container equipped with a vented lid containing 5 L of grape must. Each must sample underwent fermentation at 15 °C without any shaking for 7 days until the fermentation process was completed. The obtained wines were filter-sterilized, bottled, and stored at 4 °C. Further analysis and sensory evaluation were conducted after a 2-month storage period.

### 2.3. Quality Characteristics of Korean White Wine

All wine samples underwent centrifugation (3578× *g*, 10 min) for fermentation characteristic analysis. Soluble solids were measured using a refractometer, and the reducing sugar content was determined using dinitrosalicylic acid according to AOAC guidelines [25]. The pH was measured using a pH meter (Mettler-Toledo, Schwerzenbach, Switzerland), and the total acidity was determined via titration of filtrates with 0.1 N NaOH (expressed as g/L of tartaric acid) [26]. The alcohol content was measured using a hydrometer based on the specific gravity of wine distillates (expressed as % *v*/*v*) at 15 °C [25]. Hue and intensity values were derived from OD_420_/OD_520_ and OD_420_ + OD_520_ [27]. Color values were measured using a colorimeter (CM-3600d, Minolta Co., Osaka, Japan) calibrated with a standard white plate (L = 97.80, a = −0.38, b = 2.03) [28]. Free sugar and organic acid concentrations were analyzed by using HPLC (Model Prominence, Shimadzu, Kyoto, Japan) with a Sugar-Pak I column (diameter 6.5 × 300 mm; Waters, Milford, MA, USA) and a PL Hi-Plex H column (diameter 7.7 × 300 mm; Agilent Technologies, Santa Clara, CA, USA). Chromatography conditions for free sugars were as follows: flow rate of 0.5 mL/min, temperature of 90 °C, and mobile phase of 50 mg/L Ca–ethylenediaminetetraacetic acid (Ca–EDTA) buffer [29]. For the organic acids, conditions included a flow rate of 0.6 mL/min, temperature of 65 °C, and mobile phase of 0.005 mol sulfuric acid. Free sugars and organic acids were detected using a refractive index detector (RID-10A, Shimadzu) [30].

### 2.4. Total Phenolic Compounds in Korean White Wine

The total phenolic compound content was assessed using the Folin–Ciocalteu method [31]. Briefly, 2 mL of wine samples were combined with 2 mL of 1:1 (*v*/*v*) Folin–Ciocalteu reagent and incubated at room temperature for 3 min. Subsequently, each tube received an addition of 2 mL of 10% Na_2_CO_3_, was vortexed, and was allowed to stand at room temperature for 1 h. Absorbance was measured at 700 nm, and the outcomes were expressed as gallic acid equivalents in mg/L of Korean white wine.

### 2.5. Total Flavonoid Content in Korean White Wine

The total flavonoid content of the Korean white wine samples was assessed following the methodology outlined by Choi et al. (2020) [32]. The wine samples were examined using a spectrophotometer (UV-1601, Shimadzu Co., Kyoto, Japan) at 510 nm, with comparison against a blank solution containing all reagents along with 200 µL of distilled water in lieu of wine samples. In the procedure, 430 µL of 50% ethanol, 70 µL of a wine sample, and 50 µL of 5% NaNO_2_ were mixed in a test tube. After a 30 min incubation, the samples were mixed with 50 µL of 10% Al(NO_3_)_3_·9H_2_O. Following a 6 min interval, 500 µL of NaOH (1 N) was added, and the solutions were vortexed for 5 s. Results were expressed as rutin equivalents in mg/L of Korean white wine.

### 2.6. Evaluation of Antioxidant Activities of Korean White Wine

The diphenylpicrylhydrazyl (DPPH) radical scavenging activity was assessed based on a method described by Oszmiański et al. (2011) [33], with minor modifications. Briefly, 100 µmol DPPH was dissolved in 96% ethanol to prepare the radical stock solution immediately prior to experimentation. Subsequently, 1 mL of DPPH was mixed with 1 mL of the wine sample and 3 mL of 96% ethanol. After thorough agitation, the mixture was stored at room temperature under dark conditions for 30 min. The decrease in the absorbance of the resulting solution was monitored at 517 nm after 30 min. The obtained results were corrected for dilution and expressed in µM Trolox/mL of Korean white wine. The absorbance was measured using a spectrophotometer (UV-1601, Shimadzu Co.).

The 2,2′-azino-bis-3-ethylbenzothiazoline-6-sulfonic acid (ABTS) radical scavenging activity was evaluated according to a method described by Oszmiański et al. (2011) [33]. Briefly, ABTS was dissolved in water to prepare a 7 µM concentration. The ABTS radical cation (ABTS) was generated via the reaction of ABTS stock solution with 2.45 µM potassium persulfate (final concentration) and was stored at room temperature under dark conditions for 12–16 h prior to utilization. This radical remained stable in this state for >2 days. Samples containing the ABTS solution were diluted with redistilled water to reach an absorbance of 0.700 ± 0.02 at 734 nm and were equilibrated at 30 °C. Subsequently, after adding 3.0 mL of diluted ABTS solution (A_734nm_ = 0.700 ± 0.02) to 30 µL of a wine sample, the absorbance was measured exactly 6 min after initial mixing. The results were corrected for dilution and expressed in µM Trolox/mL of Korean white wine. The absorbance was measured using a spectrophotometer (UV-1601, Shimadzu Co.).

The ferric ion-reducing antioxidant power (FRAP) was measured according to a method described by Oszmiański et al. (2011) [33]. This assay was based on the reducing power of a compound (an antioxidant). A potential antioxidant can reduce ferric ions (Fe^3+^) to ferrous ions (Fe^2+^), whereby the latter forms a blue complex (Fe^2+^/TPTZ), leading to an increase in absorbance at 593 nm. The FRAP reagent was prepared by mixing 300 µM acetate buffer (pH 3.6), 10 µM TPTZ solution in 40 µM HCl, and 20 µM FeCl in a ratio of 10:1:1 (*v*/*v*/*v*). Then, 300 µL of FRAP reagent and 10 µL of wine samples were added to 96-well plates and thoroughly mixed. The absorbance was measured at 593 nm after 10 min using a spectrophotometer (UV-1601, Shimadzu Co.). A standard curve was constructed using different Trolox concentrations. All solutions were prepared on the same day of the experiment. The results were corrected for dilution and expressed in µM Trolox/mL of Korean white wine.

### 2.7. Volatile Aromatic Compounds in Korean White Wine

To compare the aromatic profiles of white wine across various white grape cultivars, volatile aromatic compounds were quantified utilizing gas chromatography–mass spectrometry (7890A GC–MS; Agilent, Santa Clara, CA, USA) equipped with a flame ionization detector. Separation was carried out using a DB-WAX column (60 m × 250 µm × 0.25 mm; Waters, Milford, MA, USA), and the compounds were detected using a triple-axis Agilent 5975C Inert XL MSD detector. Helium was used as a carrier gas at a consistent flow rate of 1 mL/min. The chromatograph oven was programmed as follows: an initial hold at 40 °C for 2 min, increased at a rate of 2 °C/min up to 220 °C, and then continuously elevated at 20 °C/min until 240 °C, with a final hold at 240 °C for 5 min. Volatile aromatic compounds were extracted from the wine via solid-phase microextraction (SPME) utilizing a fiber (50/30 µm DVB/CAR/PDMS; Supelco, Bellefonte, PA, USA) in headspace (HS) mode with magnetic stirring. Subsequently, 5 mL of the sample was introduced into an HS vial (20 mm, PTFE/silicon septum, magnetic cap), accompanied by the addition of 1.25 g of NaCl to enhance the concentration of volatile aromatic compounds in the HS by increasing water-soluble component retention. Before extraction, the sample was agitated in a water bath at 35 °C for 20 min to attain equilibrium. The SPME fiber was then spiked into the vial and exposed at 30 °C for 40 min. The quantification of commercial standards was conducted using standards provided by Sigma-Aldrich (St. Louis, MO, USA). Identification of volatile aromatic compounds was based on comparison of their gas chromatograph retention times and mass spectra with reference to spectral data from the Wiley9Nist 0.8 library (Wiley9Nist 0.8 library, mass spectral search program, v. 5.0, USA) [26]. The quantity of each compound in the wine was calculated using the peak area in relation to chemical standards.

### 2.8. Sensory Evaluation

Sensory evaluation was conducted 2 months after bottling, employing a seven-point hedonic scale. Prior to evaluation, each wine was placed in a sample bottle and allowed to settle undisturbed for an hour at room temperature with the lid sealed. Subsequent to flavor assessment, the wines were poured into wine glasses for evaluation of color, taste, and overall preference. The panel was comprised of 20 judges, who are sensitive to taste discrimination and have several experiences, selected from the Department of Food Science and Technology at Kyungpook National University, Korea. The judges evaluated the wines, maintaining a minimum 3 min interval between samples, and were offered water to cleanse their palates. Sensory scores were assigned as follows: 7 (like extremely), 4 (neither like or dislike), and 1 (dislike extremely).

### 2.9. Statistical Analysis

All experiments were conducted in at least triplicate. Data were expressed as the mean and standard deviation values and analyzed using Statistical Package for the Social Sciences (SPSS, v. 12.0 for Windows). Significance was determined at *p* < 0.05 using Student’s *t*-test and one-way analysis of variance, followed by Duncan’s multiple range test. Principal component analysis (PCA) (Appendix A) was performed using GraphPad Prism (version 10.0.1 for Windows, GraphPad Software, San Diego, CA, USA; www.graphpad.com (accessed on 1 June 2023)).

## 3. Results and Discussion

### 3.1. Physicochemical Characteristics of Grape Must and Wine

In this study, the physicochemical characteristics of six types of white grape cultivars, including one domestic (Cheongsoo) and five international cultivars were analyzed (Table 1). Among these, the KN grape displayed the highest pH, whereas the VB grape showed the lowest pH; the other cultivars exhibited minor variations. Total acidity was the highest in the CB grape and the lowest in the KN grape; further, despite its low pH, the total acidity of the VB grape was not considerably high. All grape cultivars exhibited a high soluble solid content of ≥18°Brix, with the CB, SM, KN, and VB cultivars showing a higher sugar content (20°Brix) than that in the typical Korean grape cultivar Campbell Early (approximately 14°Brix). The contents of both reducing sugar and free sugar, including glucose and fructose, were notably elevated, similar to the trend observed in soluble solids, indicating that all grape cultivars had sufficient sugar content for winemaking, thereby avoiding chaptalization. Regarding the organic acid content, the VB cultivar, which displayed the lowest pH, exhibited significantly higher tartaric acid content than other cultivars. Additionally, the CB, SM, and SB cultivars showed higher citric acid levels than other cultivars. The malic acid content was the highest in the CB cultivar, whereas the KN, VB, and CS cultivars exhibited relatively lower levels.

The six types of white grape cultivars were subjected to fermentation at 15 °C, and their physicochemical characteristics were subsequently analyzed (Table 2). The VB wine, boasting the lowest initial pH among the grape cultivars, exhibited a further pH decrease to 3.19 after fermentation. Conversely, the KN wine, with the highest initial pH, also exhibited a pH decrease post-fermentation, aligning its pH with that of the CB and SB wines. The pH values of the SM and CS wines also exhibited slight reductions after fermentation. The total acidity of all the wines increased after fermentation, which was attributed to organic acid production during yeast metabolism throughout the fermentation process. Soluble solid content across the wines ranged from 6.2°Brix to 7.2°Brix, and all the wines exhibited reducing sugar levels below 1%, confirming the successful completion of fermentation. Regarding alcohol content, the CB, SM, KN, and VB wines, initially featuring sugar content of >20°Brix, exhibited alcohol levels of 11.8–12.6%, typical for wines. The SB and CS wines exhibited slightly lower alcohol content at 10.4% and 10.6%, respectively. Analysis of free sugar content indicated significant consumption of glucose and fructose, similar to the results from the reducing sugar content analysis. Concerning organic acid content, citric acid levels significantly increased in the CB and SM wines, while showing slight increments in the other wines. Moreover, tartaric acid content significantly increased in all the wines except for the KN wine, while the malic acid content remained relatively stable across all the wines. Incremental changes were observed in succinic and acetic acid contents across all the wines compared to their grape counterparts.

pH and total acidity are considered important factors that affect various sensory qualities of wine. They are influenced by the balance between the anionic forms of organic acids (primarily tartaric and malic acids) and major positive ions, primarily K [34]. Organic acids commonly derived from natural grapes include malic acid, citric acid, and tartaric acid; these grapes also contain short-chain organic acids such as succinic acid, acetic acid, and lactic acid, which are formed during various fermentation processes [35]. These organic acids considerably affect the perceived acidity, a crucial factor determining organoleptic quality. Additionally, short-chain organic acids can act as secondary supporters for iron absorption [35,36]. Factors such as grape ripening and temperature during growth can affect the tartaric acid content, and the malic acid content tends to decrease in warm cultivation regions; both acids display relative stability compared with other organic acids [13]. During the aging process, the organic acid content tends to decline due to enzymatic or chemical esterification, wherein they bind to alcohol to form esters, thereby contributing to the enhanced aromatic properties of the wine [37].

### 3.2. Total Phenolic and Flavonoid Contents in Korean White Wine

Flavonoids and phenolic compounds, which are recognized as functional components in wine, are derived from grape seeds and skins [38]. During white wine production, grape skins are not utilized during the fermentation process. Due to the relatively lower anthocyanin content in white grape skins, white wines generally exhibit reduced levels of total phenolic and flavonoid compounds compared with red wines. However, these two compound categories remain valuable indicators for exploring differences in functional characteristics among diverse grape cultivars. Therefore, the levels of total phenolic compounds and flavonoids in white wines derived from six types of white grape cultivars were analyzed (Figure 2). The results revealed significantly higher levels of total phenolic and flavonoid compounds in the VB wine, whereas the other wines expectedly showed lower levels of these compounds. This disparity could be attributed to the presence of slightly more color pigments within the skin of VB grapes (Figure 1). This facilitates higher extraction of phenolic and flavonoid compounds during the juice extraction process compared with other grape cultivars.

Phenolic compounds including flavonoids (e.g., anthocyanins, flavan-3-ols, and flavonols) and nonflavonoids (e.g., phenolic acids and stilbenes) represent secondary metabolites found in grapes and wine. They assume a pivotal role as quality indicators in wine assessment [39]. Phenolic acids, including hydroxybenzoic acid and hydroxycinnamic acid, are abundantly present in white wine. Their role as oxidative substrates and precursors contributes to the bitter taste inherent to white wine [40,41]. These compounds play a crucial role in defining attributes of white wine [39]. Phenolic compounds are predominantly present in the skin, seed, and stem of grapes, and in the case of white wine, unlike red wine, they remain largely unchanged due to the absence of a maceration process during winemaking [42]. Consequently, phenolic compounds in white wine have garnered less attention in comparison to those in red wine. Nonetheless, their potential to yield intriguing results persists, as exemplified by the case of the VB wine in our study. In fact, a study by Luo et al. [43] identified 35 phenolic compounds in Vidal pomace, with quercetin derivatives like quercetin 3-O-glucuronide and quercetin 3-O-glucoside emerging as major flavonols.

### 3.3. Antioxidant Activities of Korean White Wine

Red wine is known to possess a strong antioxidant capacity, which is attributed to the presence of various polyphenolic compounds originating from grape skins and seeds [44,45]. This property serves as a valuable indicator for evaluating the functional quality of wines. In our study, the antioxidant capacity of white wine from each grape cultivar was evaluated based on DPPH, ABTS, and FRAP activities (Figure 3); however, it was not as potent as that in red wine. The VB wine exhibited significantly higher DPPH, ABTS, and FRAP activities than other wines, consistent with the findings of the analysis of total phenolic and flavonoid compounds. Soleas et al. (1997) [46] revealed that polyphenolic compounds influencing antioxidant activity were more abundant in Vidal grapes than in Chardonnay and Seibel grapes but less abundant in Riesling grapes. Additionally, Nile et al. (2015) compared antioxidant activities of 20 different grape cultivars and demonstrated that red grape cultivars, such as Chasselas Rouge, Red Globe, Delaware, Ruby Seedless, Koho, Hongiseul, and Honey Red, exhibited DPPH activity levels ranging from 48.2 to 87.6 µg/100 g, FRAP activity levels ranging from 115.0 to 179.0 µg/100 g, and ABTS activity levels ranging from 11.1 to 39.0 µg/100 g [47]. In contrast, Vidal grapes displayed relatively lower DPPH, FRAP, and ABTS activity of 34.5, 86.7, and 14.9 µg/100 g, respectively. However, the Vidal cultivar showed higher antioxidant activity than the commonly cultivated Campbell Early in Korea, which displayed DPPH and FRAP activity levels of 32.8 µg/100 g and 79.1 µg/100 g, respectively [47]. This suggests that Vidal grapes exhibit a moderate level of antioxidant capacity among white wines. Our study results highlight that VB, enriched with high polyphenolic content relative to other white grape cultivars, demonstrated the most pronounced antioxidant activity.

Polyphenols and sulfur compounds, which contribute to the antioxidant capacity of wine, are typically more abundant in red wine than in white wine. Nonetheless, recent research has revealed that the antioxidant capacity of Chardonnay wine can be improved by adjusting the contents of amino acids, aromatic compounds, and peptides containing N- and S-compounds through glutathione treatment in the early stages of winemaking [48]. Furthermore, a previous study revealed the role of sulfur compounds in augmenting the radical quenching ability of white wine [49], highlighting the increasing interest in the wine metabolome. As the mechanisms of antioxidant substances can be classified into two categories—ROS scavengers and transition metal ion chelators, various methods such as ORAC, TRAP, FRAP, CUPRAC, ABTS, and DPPH have been employed to measure antioxidant capacity [50]. The presence of a greater number of procyanidin trimers in Vidal grapes than in other grape cultivars indicate their abundance in phenolic compounds [43], potentially accounting for the relatively higher antioxidant capacity in the VB wine compared with that in other wines in this study.

### 3.4. Color Differences between Korean White Wines

The color of wine is a key sensory indicator, subject to influence from grape composition, vinification techniques, and storage conditions [51]. The hue, intensity, and Hunter’s color values of white wines derived from six types of white grape cultivars were determined (Table 3). Regarding hue, the VB wine exhibited the lowest value, whereas the CS wine had the highest value. As for intensity, the VB wine displayed notably greater intensity compared to the other wines, with the SB wine showing the lowest intensity. Hunter’s color value analysis showed that the VB wine exhibited the lowest L value, whereas the remaining wines showed no significant variations. In terms of a and b values, the VB wine, known for its slightly redder grape skins, registered higher values than the other wines. The a values of the other wines showed no large differences, whereas the b values of the CS and SM wines slightly exceeded those of the other wines, except for the VB wine.

### 3.5. Volatile Aromatic Compounds in Korean White Wines

Even under similar climatic, soil, and environmental conditions, the chemical composition of grapes can vary significantly across different grape cultivars [52]. Diverse volatile aromatic compounds, such as esters, higher alcohols, monoterpenes, and acids, can be produced during the fermentation process owing to the presence of various volatile aromatic compounds (precursors) in different grape cultivars, all of which play a vital role in determining wine quality [34,53]. In the six different white grape cultivars examined in this study, the distribution of volatile aromatic compounds differed significantly after alcohol fermentation. Through GC–MS analysis, 2 acetals, 3 acids, 4 aldehydes, 16 higher alcohols, 30 esters, and 8 monoterpenes were identified (Table 4). The acetal group, which is abundantly found in distilled alcoholic beverages, contributes to pleasant aromas such as apple, pineapple, cherry, and fruity aromas [54]. Among the six white wines, the SM and KN wines exhibited higher levels of the acetal group than the other wines. Acids also contribute to wine aroma. Short-chain acids such as acetic, propanoic, and butanoic acids are formed as metabolites of alcoholic fermentation, whereas medium-chain acids such as hexanoic, octanoic, decanoic, and dodecanoic acids are possibly formed as intermediates in long-chain fatty acid biosynthesis [55]. In the current study, three long-chain acids, namely, heptanoic, octanoic, and decanoic acids, were detected, with distinctive distribution patterns in each wine. Low levels of heptanoic acid were detected across all the wines, whereas high levels of octanoic acid were found in the SB and CS wines. Conversely, higher levels of decanoic acid were observed in the CB, KN, and CS wines than in other wines. Interestingly, the CS wine had significantly higher levels of both octanoic and decanoic acids, whereas other wines showed higher levels of only one of these acids. Acetaldehyde, an intermediate product in the formation of ethanol and glycerol during fermentation, can impart an off flavor when present in high concentrations [56]. However, the wines examined in this study contained low levels of acetaldehyde (<5.05 μg/mL). Similarly, other aldehydes were detected in low levels, indicating that the variation in their levels among grape cultivars was relatively small compared with that of other volatile compounds. Higher alcohols constitute the largest amount of volatile aromatic compounds in wine and serve as crucial precursors for ester formation [34]. The higher alcohol content in each wine varied significantly among grape cultivars in this study, with isoamyl alcohol and 2-phenylethanol identified as the primary components. These two compounds were most prominently detected in the SM and KN wines, whereas the SB wine showed the lowest levels. Although notable variations were observed in the concentrations of other higher alcohols across different wines, the total content of higher alcohols was largely determined by isoamyl alcohol and 2-phenylethanol. These two compounds were also detected in high levels in white wines prepared using autochthonous grape cultivars, Krstač and Žižak [57]. The total content of higher alcohols in the CB, VB, and CS wines was comparable, but the SB wine showed the lowest levels. Esters are known to considerably influence wine aroma. In this study, several esters, including ethyl acetate, isoamyl acetate, ethyl hexanoate, ethyl octanoate, ethyl decanoate, ethyl 9-decenoate, and 2-phenylethyl acetate, were detected in significantly higher concentrations than other ester compounds across all wines. These compounds strongly influence the aroma of white wine. The concentrations of these ester compounds directly contributed to the total ester content, which was remarkably higher in CB, SB, and CS wines than in the other wines, whereas the SM wine showed the lowest total ester content. Further, significant variations were observed in the concentrations of isobutyl acetate and hexyl acetate among the different grape cultivars. Interestingly, 2-phenylethyl acetate, despite originating from its precursor, 2-phenylethanol, which was present in higher amounts in the SM and KN wines, exhibited lower concentrations in these wines. This phenomenon can be attributed to the enzymatic activity of alcohol acetyltransferase during fermentation, a finding consistent with previous research [58]. A study conducted by Kim et al. (2016) focused on the volatile compounds present in white wines produced from the grape cultivars Cheongsoo, Chardonnay, and Riesling cultivated in Korea. That study reported that the Cheongsoo wine exhibited a total ester content that was more than four times higher than that of Chardonnay and Riesling wines. Notably, the concentrations of compounds such as ethyl acetate, isoamyl acetate, hexyl acetate, ethyl hexanoate, ethyl octanoate, and ethyl decanoate were prominently elevated in the Cheongsoo wine compared to the other two cultivars [19]. Different groups of volatile aromatic compounds exist in wine at varying concentrations, and their impact on the sensory characteristics of wine can differ based on their odor thresholds [59,60]. Monoterpene compounds for instance, are well known for imparting floral characteristics to wine and are commonly found in various white grape cultivars such as Muscat, Gewürtztramminer, Riesling, Sylaner, Traminer, Hüxel, and Müller Thurgau [61]. These compounds possess low odor thresholds (100–400 ppb), making even minor concentration changes influential in shaping the aroma profile of the wine [32,62]. In the present study, the relative proportions of monoterpene compounds in the wines produced from different grape cultivars exhibited the most significant variation compared to other groups of volatile aromatic compounds. Notably, the wine produced from the CS grape showed a higher diversity of monoterpene compounds, with the detection of eight different types, surpassing the other wines. In particular, the CS wine displayed substantial levels of 4-terpineol and p- cymene, along with significantly higher levels of β-citronellol and geraniol than the other wines. Interestingly, linalool, a compound commonly found in various white wines, was detected in exceptionally high levels in the KN wine when compared to the other wines. The total content of monoterpene compounds in the CS wine was 1.4–10 times higher than that in the other wines, underscoring the significant influence of the monoterpene-based floral aromatic profile on the sensory attributes of the CS wine in comparison to the other wines. 

PCA was conducted on the volatile ester and monoterpene compounds, which have a substantial impact on wine aroma, in order to compare the relationships between different wine cultivars and their respective groups based on variable loadings and PC scores (Figure 4). In the analysis of volatile ester compounds, PC1 and PC2 explained 44.59% and 25.37% of the variance, respectively. Several ester compounds were closely associated with the CB, SB, and CS wines, which exhibited higher total ester contents compared to the other wines. Specifically, ethyl acetate, propyl acetate, isobutyl acetate, ethyl butanoate, isoamyl acetate, ethyl hexanoate, hexyl acetate, ethyl octanoate, isobutyl octanoate, isobutyl decanoate, methyl salicylate, and 2-phenylethyl acetate were significant contributors to the SB and CS wines. The CB wine was notably influenced by isoamyl hexanoate, ethyl nonanoate, ethyl decanoate, isopentyl octanoate, and ethyl 9-hexadecenoate. Ethyl heptanoate and ethyl 4-hydroxybutanoate had an impact on the SM wine, while ethyl isobutyrate and ethyl 9-decenoate were relevant to the VB wine. Lastly, the KN wine demonstrated associations with ethyl propanoate, ethyl 2-hexenoate, and ethyl hexadecanoate.

In the analysis of monoterpene compounds, PC1 and PC2 explained 40.56% and 33.94% of the variance, respectively. The loadings were found to be closely associated with the CS and KN wines, which contained higher levels of monoterpene compounds. This suggests that volatile monoterpene compounds played a significant role in influencing the sensory characteristics of these two wines compared to the other wines. Specifically, p-cymene, 4-terpineol, β-citronellol, and geraniol were significant contributors to the aroma profile of the CS wine. However, terpinolene, nerol, linalool, and α-terpineol were influential in shaping the aroma of the KN wine. The other wines were not significantly impacted by monoterpenes, based on the PCA analysis.

### 3.6. Sensory Evaluation of Korean White Wine

Based on the examined grape cultivars, our analysis revealed significant variations in the physicochemical properties of volatile aromatic compounds of the wines. These differences may be attributed to the complex interactions and changes in the diverse chemical components present in raw grapes, which contribute to distinct sensory profiles. Figure 5 shows the sensory scores for color, flavor, taste, acidity, and the overall preference of each white wine. No statistically significant difference was observed in any category, except for flavor, and the CS wine exhibited the highest scores for color, flavor, taste, and overall preference. In terms of color evaluation, despite having the least differences among the wines, the VB wine, which exhibited a low hue value, high intensity, and elevated a and b values, showed the lowest score. In the flavor evaluation, the CS wine, which contained the highest levels of monoterpene compounds and was most influenced by volatile esters and terpenes (as determined via PCA analysis), exhibited the highest flavor score. In contrast, the VB wine, with less influence from volatile compounds (as determined via PCA analysis), showed the lowest flavor score. In terms of taste evaluation, the CS wine, with a relatively lower alcohol content, showed the highest score, whereas the SB wine showed a comparable alcohol content. This outcome reflects the complex interplay of sweetness, acidity, and alcohol in determining taste perception. Acidity evaluation revealed that the SM wine, with the highest acetic acid content, showed the lowest score in acidity. Meanwhile, the other wines showed similar scores in this category. Regarding overall preference, the CS wine, which displayed strong scores in color, flavor, and taste, exhibited the highest score. The VB wine, despite lower scores in color and flavor but higher scores in taste and acidity, showed the second-highest score in overall preference.

## 4. Conclusions

Cultivating European white grape cultivars suitable for winemaking in Korea has been challenging due to factors such as climate, soil conditions, precipitation, and sunlight. However, recent advancements in cultivation techniques have led to the successful cultivation of these cultivars in Korea. In this study, the newly bred Korean white grape cultivar Cheongsoo was a focus, and its physicochemical properties were investigated. Moreover, six types of white wines from these grapes were produced and their characteristics were evaluated to confirm their suitability for winemaking. All six grape cultivars demonstrated high sugar content of >18.2°Brix, indicating their potential for winemaking without the need for chaptalization. Despite being grown in the same vineyard, distinct variations in chemical components were observed among the grape cultivars. Notably, significant differences were noted in antioxidant capacity, color, and the content of various volatile aromatic compounds. The VB wine exhibited enhanced antioxidant capacity, whereas the CS wine showed the presence of a diverse range of monoterpene compounds. Moreover, in the sensory evaluations, both wines showed higher overall preference scores than other wines. These findings provide valuable insights into the grape wine industry in Korea, serving as fundamental resources for diverse experimentation and exploration of different grape cultivars and their characteristics.

## Figures and Tables

**Figure 1 foods-12-03246-f001:**
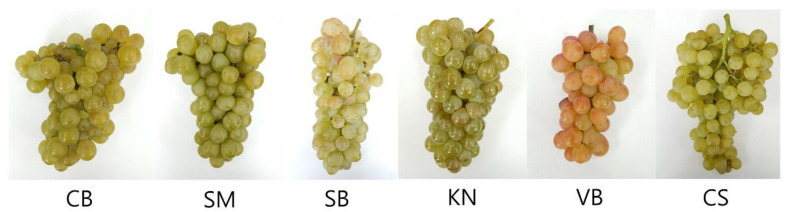
Six types of white grape cultivars used in this study.

**Figure 2 foods-12-03246-f002:**
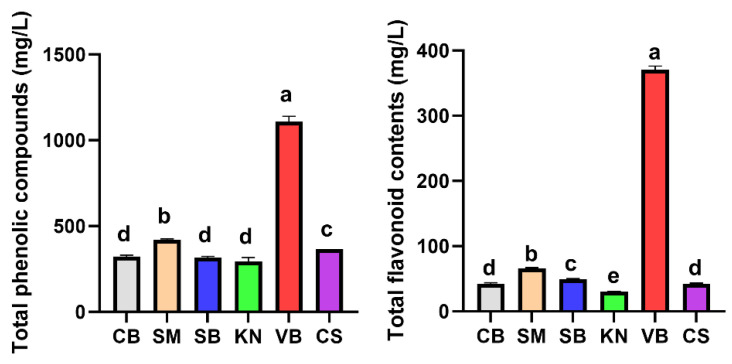
Total phenolic and flavonoid contents in white wines produced using six types of white grapes cultivated in Korea. Different letters within the same column indicate significant differences (*p* < 0.05).

**Figure 3 foods-12-03246-f003:**
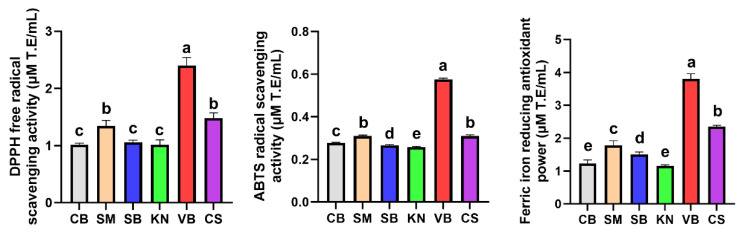
Antioxidant activities of white wines produced using six types of white grapes cultivated in Korea. Different letters within the same column indicate significant differences (*p* < 0.05).

**Figure 4 foods-12-03246-f004:**
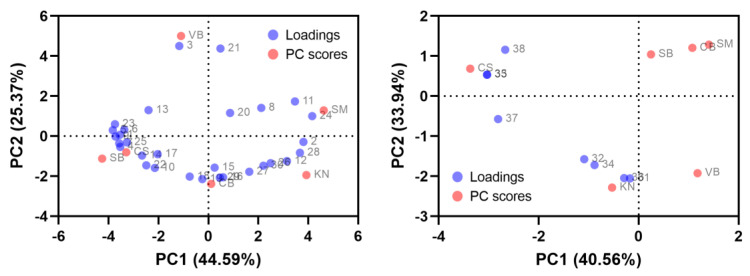
Biplots of variable loadings and principal component scores from the principal component analysis (PCA) of volatile ester (**left**, 1–30) and monoterpene (**right**, 31–38) compounds. Numbers represent the following compounds: 1, ethyl acetate; 2, ethyl propanoate; 3, ethyl isobutyrate; 4, propyl acetate; 5, isobutyl acetate; 6, ethyl butanoate; 7, isoamyl acetate; 8, isopentyl propanoate; 9, ethyl hexanoate; 10, hexyl acetate; 11, ethyl heptanoate; 12, ethyl 2-hexenoate; 13, heptyl acetate; 14, ethyl octanoate; 15, isoamyl hexanoate; 16, ethyl nonanoate; 17, isobutyl octanoate; 18, ethyl decanoate; 19, isopentyl octanoate; 20, diethyl succinate; 21, ethyl 9-decenoate; 22, isobutyl decanoate; 23, methyl salicylate; 24, ethyl 4-hydroxybutanoate; 25, 2-phenylethyl acetate; 26, ethyl dodecanoate; 27, isoamyl decanoate; 28, ethyl hexadecanoate; 29, ethyl 9-hexadecenoate; 30, ethyl octadecenoate; 31, terpinolene; 32, nerol; 33, p-cymene; 34, linalool; 35, 4-terpineol; 36, α-terpineol; 37, β-citronellol; 38, geraniol.

**Figure 5 foods-12-03246-f005:**
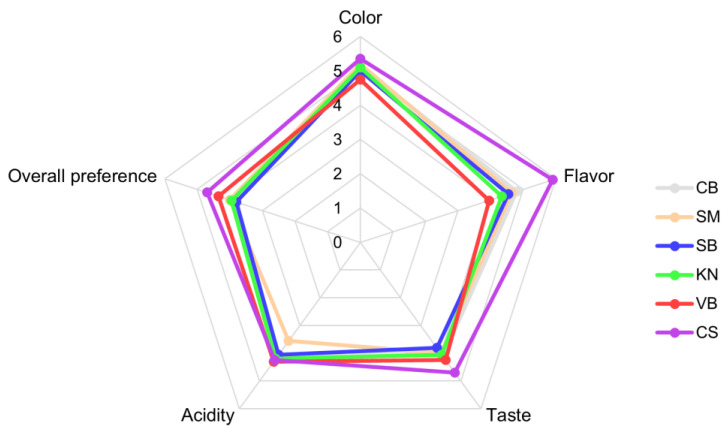
Sensory scores of white wines produced using six types of white grape cultivars grown in Korea.

**Table 1 foods-12-03246-t001:** Physicochemical characteristics of six types of Korean white grape must.

Item	CB	SM	SB	KN	VB	CS
pH	3.80 ± 0.02 b	3.70 ± 0.03 c	3.83 ± 0.02 b	4.35 ± 0.05 a	3.39 ± 0.03 d	3.63 ± 0.03 c
Total acidity (g/L)	6.0 ± 0.0 a	4.4 ± 0.0 c	5.1 ± 0.0 b	2.3 ± 0.0 f	4.0 ± 0.0 d	3.6 ± 0.0 e
Soluble solid (°Brix)	20.7 ± 0.2 c	22.0 ± 0.1 b	18.2 ± 0.2 e	22.6 ± 0.2 a	21.0 ± 0.1 c	18.8 ± 0.1 d
Reducing sugar (%)	22.29 ± 0.46 bc	23.65 ± 0.10 a	19.08 ± 0.41 d	22.68 ± 0.38 b	21.72 ± 0.53 c	19.31 ± 0.59 d
Free sugar (mg/mL)						
Glucose	108.68 ± 1.51 b	115.45 ± 2.03 a	92.64 ± 0.96 d	109.56 ± 2.68 b	102.71 ± 1.84 c	95.98 ± 2.15 d
Galactose	ND	ND	ND	ND	ND	ND
Fructose	108.16 ± 0.91 d	119.41 ± 1.30 b	97.55 ± 1.62 f	112.56 ± 1.18 c	128.38 ± 0.94 a	101.48 ± 1.35 e
Organic acid (mg/mL)						
Citric acid	0.68 ± 0.04 a	0.71 ± 0.03 a	0.51 ± 0.02 b	0.22 ± 0.03 d	0.39 ± 0.03 c	0.39 ± 0.04 c
Tartaric acid	2.69 ± 0.18 b	2.82 ± 0.24 b	2.04 ± 0.15 c	1.60 ± 0.18 d	4.13 ± 0.28 a	3.02 ± 0.23 b
Malic acid	9.46 ± 0.85 a	7.82 ± 0.77 b	6.01 ± 0.73 c	4.64 ± 0.57 d	4.28 ± 0.34 d	3.35 ± 0.43 e
Succinic acid	0.03 ± 0.00	ND	ND	ND	ND	ND
Acetic acid	ND	ND	ND	ND	0.02 ± 0.01	ND

Different letters within the same column indicate significant difference (*p* < 0.05). ND, not detected.

**Table 2 foods-12-03246-t002:** Physicochemical characteristics of Korean white wines produced using six types of white grape cultivars.

Item	CB	SM	SB	KN	VB	CS
pH	3.79 ± 0.02 b	3.59 ± 0.01 d	3.75 ± 0.02 c	3.84 ± 0.03 a	3.19 ± 0.02 f	3.40 ± 0.03 e
Total acidity (g/L)	8.0 ± 0.0 a	7.7 ± 0.0 b	6.3 ± 0.0 d	5.7 ± 0.0 e	6.6 ± 0.0 c	5.3 ± 0.0 f
Soluble solid (°Brix)	7.0 ± 0.0 b	7.0 ± 0.1 b	6.2 ± 0.1 d	7.0 ± 0.0 b	6.8 ± 0.1 c	7.2 ± 0.1 a
Reducing sugar (%)	0.10 ± 0.01 d	0.30 ± 0.02 b	0.12 ± 0.02 d	0.53 ± 0.02 a	0.26 ± 0.01 c	0.11 ± 0.01 d
Alcohol (%)	11.8 ± 0.1 d	12.2 ± 0.1 c	10.4 ± 0.0 f	12.4 ± 0.1 b	12.6 ± 0.1 a	10.6 ± 0.0 e
Free sugar (mg/mL)						
Glucose	ND	ND	ND	0.34 ± 0.04 a	ND	0.38 ± 0.06 a
Galactose	0.15 ± 0.01 b	0.17 ± 0.02 ab	ND	ND	0.21 ± 0.02 a	0.12 ± 0.01 c
Fructose	0.46 ± 0.06 c	0.78 ± 0.08 b	0.32 ± 0.05 c	1.03 ± 0.12 a	0.65 ± 0.09 b	0.40 ± 0.06 c
Organic acid (mg/mL)						
Citric acid	1.27 ± 0.20 a	1.17 ± 0.13 a	0.61 ± 0.10 b	0.42 ± 0.06 b	0.48 ± 0.07 b	0.46 ± 0.04 b
Tartaric acid	2.78 ± 0.19 b	2.81 ± 0.23 b	2.10 ± 0.17 c	1.58 ± 0.21 d	4.07 ± 0.35 a	3.09 ± 0.24 b
Malic acid	8.71 ± 0.68 a	7.68 ± 0.82 a	5.32 ± 0.49 b	4.93 ± 0.58 b	3.37 ± 0.38 c	2.95 ± 0.41 c
Succinic acid	2.22 ± 0.16 a	2.09 ± 0.22 a	0.99 ± 0.13 c	1.93 ± 0.15 a	1.37 ± 0.16 b	0.96 ± 0.10 c
Acetic acid	0.03 ± 0.01 c	0.17 ± 0.02 a	0.07 ± 0.01 c	0.12 ± 0.03 b	0.05 ± 0.01 c	0.04 ± 0.01 c

Different letters within the same column indicate significant difference (*p* < 0.05). ND, not detected.

**Table 3 foods-12-03246-t003:** Hue, intensity, and Hunter’s color values of white wines produced using six types of white grape cultivars grown in Korea.

Item	CB	SM	SB	KN	VB	CS
Hue	3.26 ± 0.04 e	3.91 ± 0.02 b	3.50 ± 0.10 d	3.78 ± 0.09 c	2.53 ± 0.01 f	4.74 ± 0.06 a
Intensity	0.16 ± 0.01 c	0.17 ± 0.01 c	0.11 ± 0.00 d	0.16 ± 0.01 c	0.38 ± 0.02 a	0.21 ± 0.01 b
Hunter’s color values						
L	61.24 ± 0.11 a	60.69 ± 0.23 b	61.03 ± 0.13 ab	60.39 ± 0.13 c	58.96 ± 0.16 d	60.90 ± 0.19 ab
a	0.19 ± 0.00 c	0.30 ± 0.01 b	0.22 ± 0.01 c	0.04 ± 0.01 d	1.70 ± 0.10 a	−0.19 ± 0.03 e
b	1.04 ± 0.09 e	2.67 ± 0.13 c	0.12 ± 0.01 f	1.44 ± 0.06 d	7.76 ± 0.25 a	4.93 ± 0.18 b

Different letters within the same column indicate significant differences (*p* < 0.05).

**Table 4 foods-12-03246-t004:** Concentrations of volatile aromatic compounds in Korean white wines produced using six types of white grape cultivars.

Compounds	Odor Description	Threshold (mg/L)	OAV	Volatile Aromatic Compounds (μg/mL)					
				CB	SM	SB	KN	VB	CS
Acetal									
1,1-diethoxyethane				65.60 ± 5.36 b	88.92 ± 7.64 a	70.11 ± 8.18 b	101.36 ± 13.05 a	47.92 ± 3.93 c	56.42 ± 4.79 bc
1-ethoxy-1-pentoxyethane				9.44 ± 1.04 b	13.52 ± 1.16 a	7.67 ± 0.83 bc	14.50 ± 1.22 a	6.61 ± 0.88 c	7.37 ± 0.67 bc
∑Acetal				75.04 ± 6.40 b	102.44 ± 8.80 a	77.78 ± 9.01 b	115.86 ± 14.27 a	54.53 ± 4.81 c	63.79 ± 5.46 bc
Acid									
Heptanoic acid	Fatty, dry	3	0.6–7	1.99 ± 0.25 c	3.97 ± 0.40 c	1.78 ± 0.32 c	9.10 ± 0.68 b	21.13 ± 2.55 a	2.03 ± 0.16 c
Octanoic acid	Rancid, cheese, fatty acid	0.5	5.6–678	7.06 ± 0.91 d	2.81 ± 0.32 d	338.98 ± 26.92 a	ND	74.87 ± 6.69 c	266.95 ± 28.89 b
Decanoic acid	Fatty, rancid	1	0.5–239	191.09 ± 16.00 b	94.49 ± 10.18 c	0.54 ± 0.15 d	173.41 ± 20.43 b	ND	239.02 ± 27.71 a
∑Acid				200.14 ± 17.16 c	101.27 ± 10.90 d	341.31 ± 27.39 b	182.50 ± 21.11 c	96.00 ± 9.24 d	508.01 ± 56.76 a
Aldehyde									
Acetaldehyde	Bitter almond	100	<1	4.15 ± 0.23 ab	5.05 ± 0.47 a	4.90 ± 0.43 a	3.36 ± 0.32 b	4.13 ± 0.28 ab	4.49 ± 0.39 a
Decanal		0.01	187–321	2.21 ± 0.18 b	2.34 ± 0.23 b	1.87 ± 0.20 b	2.38 ± 0.17 b	3.21 ± 0.41 a	ND
Dodecanal				4.46 ± 0.30 d	8.81 ± 0.67 b	6.19 ± 0.54 c	8.35 ± 0.73 b	11.31 ± 1.17 a	11.73 ± 1.32 a
4-Propylbenzaldehyde				2.66 ± 0.19 b	1.84 ± 0.14 c	2.39 ± 0.22 b	1.61 ± 0.15 c	3.17 ± 0.23 a	1.45 ± 0.21 c
∑Aldehyde				13.48 ± 0.90 c	18.04 ± 1.51 b	15.36 ± 1.39 bc	15.70 ± 1.37 bc	21.82 ± 2.09 a	17.68 ± 1.92 b
Higher alcohol									
1-Propanol	Alcohol, ripe fruity	306	<1	33.92 ± 0.38 b	22.48 ± 0.17 d	46.86 ± 0.36 a	10.95 ± 1.18 e	27.28 ± 2.67 c	36.20 ± 4.11 b
Isobutanol	Alcohol, solvent, green, bitter	75	1–1.5	80.85 ± 6.65 b	88.37 ± 8.01 b	94.27 ± 10.11 ab	73.95 ± 7.65 b	113.06 ± 12.18 a	92.80 ± 10.10 ab
1-Butanol	Medicinal, phenolic	150	<1	3.36 ± 0.41 c	5.88 ± 0.62 a	3.60 ± 0.29 bc	4.53 ± 0.50 b	5.90 ± 0.52 a	3.62 ± 0.33 bc
Isoamyl alcohol	Solvent, sweet, nail polish	60	23–47	2134.17 ± 167.85 b	2796.07 ± 226.35 a	1368.35 ± 176.28 c	2596.71 ± 206.71 a	2167.93 ± 196.88 b	1979.67 ± 185.27 b
3-Methyl-1-pentanol	Pungent, solvent, green	0.5	3–10	3.21 ± 0.28 b	4.70 ± 0.39 a	1.57 ± 0.20 c	5.13 ± 0.47 a	5.09 ± 0.49 a	2.67 ± 0.33 b
1-Hexanol	Herbaceous, grass, woody	1.1	12–37	33.94 ± 3.19 b	40.23 ± 3.58 a	13.07 ± 1.15 c	34.04 ± 2.99 b	6.63 ± 1.01 d	ND
3-Ethoxy-1-propanol	Fruity	0.1	12–57	3.75 ± 0.39 c	1.33 ± 0.15 e	5.70 ± 0.63 a	1.16 ± 0.14 e	4.60 ± 0.38 b	2.91 ± 0.25 d
1-Heptanol	Oily	0.2	14–75	9.75 ± 1.01 c	15.03 ± 1.30 a	2.85 ± 0.32 e	9.07 ± 0.83 c	12.07 ± 1.12 b	6.35 ± 0.58 d
2-Ethyl-1-hexanol		8	<1	2.92 ± 0.30 bc	2.93 ± 0.28 bc	3.66 ± 0.41 ab	2.96 ± 0.37 bc	2.29 ± 0.12 c	3.85 ± 0.33 a
2-Nonanol				ND	1.55 ± 0.09 d	12.05 ± 0.89 a	5.19 ± 0.72 b	3.84 ± 0.42 c	2.39 ± 0.28 d
levo-2,3-butanediol	Fruity, sweet, butter	150	<1	10.58 ± 1.52 bc	11.34 ± 1.35 b	8.80 ± 1.03 bc	9.12 ± 0.82 bc	14.22 ± 1.22 a	8.08 ± 0.90 c
1-Octanol	Jasmine, lemon	0.8	6–17	9.86 ± 1.05 b	5.60 ± 0.62 c	13.32 ± 1.45 a	4.94 ± 0.60 c	6.93 ± 0.58 c	11.29 ± 1.47 b
2,3-Butanediol	Floral, fruity, herbal, buttery	150	<1	2.97 ± 0.35 b	2.77 ± 0.25 b	2.07 ± 0.23 c	1.86 ± 0.17 c	3.63 ± 0.23 a	1.95 ± 0.20 c
1-Decanol	Floral, fruity, bitter, winey	0.4	5–17	3.92 ± 0.36 c	2.54 ± 0.30 d	6.65 ± 0.49 a	1.81 ± 0.13 e	3.18 ± 0.33 d	5.37 ± 0.61 b
2-Phenylethanol	Rose, honey	14	16–74	614.96 ± 52.74 b	1037.54 ± 96.83 a	219.11 ± 25.41 c	953.89 ± 88.29 a	654.97 ± 69.10 b	611.33 ± 62.47 b
Nerolidol				1.68 ± 0.20 d	2.45 ± 0.22 c	2.10 ± 0.18 cd	1.83 ± 0.19 d	5.43 ± 0.39 a	3.71 ± 0.35 b
∑Higher alcohol				2949.84 ± 236.41 b	4040.81 ± 340.51 a	1804.05 ± 219.43 c	3717.14 ± 311.76 a	3037.06 ± 287.64 b	2772.17 ± 267.58 b
Ester									
Ethyl acetate	Pineapple, fruity, balsamic	12	18–54	509.95 ± 53.60 b	225.56 ± 21.77 c	650.57 ± 58.67 a	211.36 ± 19.82 c	491.96 ± 45.63 b	448.75 ± 42.81 b
Ethyl propanoate	Fruity	1.8	3–5	8.10 ± 0.90 a	8.32 ± 0.68 a	5.84 ± 0.55 b	8.59 ± 0.78 a	7.22 ± 0.69 ab	7.20 ± 0.75 ab
Ethyl isobutyrate	Sweet, rubber			ND	1.17 ± 0.12 d	1.88 ± 0.20 c	2.22 ± 0.19 c	3.55 ± 0.38 a	2.72 ± 0.31 b
Propyl acetate	Sweet, fruity	4.7	0.4–4	8.01 ± 0.69 b	ND	16.82 ± 1.49 a	1.81 ± 0.31 d	5.90 ± 0.41 c	8.53 ± 0.67 b
Isobutyl acetate	Fruity, apple, banana	1.6	2–21	18.41 ± 1.58 d	4.03 ± 0.48 e	33.94 ± 0.36 a	2.48 ± 0.27 e	21.72 ± 2.31 c	24.42 ± 2.34 b
Ethyl butanoate	Banana, pineapple, strawberry	0.4	41–90	35.96 ± 3.67 a	16.24 ± 1.49 b	33.68 ± 3.33 a	19.55 ± 1.65 b	34.77 ± 3.19 a	33.68 ± 3.82 a
Isoamyl acetate	Banana	0.16	2500<	2024.93 ± 216.88 b	483.34 ± 53.06 d	2397.77 ± 230.87 a	413.16 ± 40.55 d	1485.98 ± 136.73 c	1799.90 ± 162.80 b
Isopentyl propanoate				3.18 ± 0.25 a	2.78 ± 0.28 b	1.27 ± 0.09 d	2.12 ± 0.15 c	2.63 ± 0.25 b	2.18 ± 0.21 c
Ethyl hexanoate	Banana, green apple	0.08	5400<	738.57 ± 70.62 a	440.66 ± 41.58 b	785.32 ± 69.11 a	435.39 ± 45.38 b	718.82 ± 70.22 a	841.88 ± 79.09 a
Hexyl acetate	Apple, cherry, pear, floral	1.5	23–118	177.36 ± 16.31 a	35.04 ± 3.19 c	122.93 ± 14.17 b	34.99 ± 3.66 c	ND	172.37 ± 14.92 a
Ethyl heptanoate	Fruit	0.22	4–30	2.48 ± 0.31 b	6.58 ± 0.59 a	0.92 ± 0.08 d	2.66 ± 0.30 b	2.31 ± 0.19 b	1.62 ± 0.15 c
Ethyl 2-hexenoate				5.75 ± 0.52 a	4.62 ± 0.48 b	1.85 ± 0.20 c	4.90 ± 0.39 b	0.99 ± 0.10 d	1.23 ± 0.11 d
Heptyl acetate				7.43 ± 0.65 a	2.22 ± 0.19 d	4.77 ± 0.44 c	1.67 ± 0.16 d	6.62 ± 0.58 b	4.81 ± 0.41 c
Ethyl octanoate	Fruity, sweet, banana, pear	0.2	7000<	2287.95 ± 230.52 a	1433.36 ± 136.77 b	2332.88 ± 250.08 a	2289.31 ± 216.38 a	2116.07 ± 199.85 a	2539.96 ± 241.71 a
Isoamyl hexanoate				3.20 ± 0.28 a	2.51 ± 0.19 bc	2.07 ± 0.18 cd	2.70 ± 0.26 b	1.81 ± 0.21 d	3.36 ± 0.35 a
Ethyl nonanoate				4.99 ± 0.43 b	3.08 ± 0.29 c	5.38 ± 0.58 b	6.29 ± 0.48 a	1.27 ± 0.17 d	3.18 ± 0.24 c
Isobutyl octanoate				3.49 ± 0.39 b	2.81 ± 0.26 b	3.56 ± 0.33 b	ND	ND	7.43 ± 0.67 a
Ethyl decanoate	Fatty acids, fruity, soap	0.2	2500<	1161.12 ± 96.70 b	865.77 ± 69.96 c	1589.28 ± 153.88 a	1468.20 ± 139.41 a	505.00 ± 47.66 d	1201.97 ± 108.60 b
Isopentyl octanoate				50.15 ± 4.58 a	28.27 ± 2.61 c	33.08 ± 2.78 bc	37.86 ± 3.59 b	2.95 ± 0.27 d	48.68 ± 3.82 a
Diethyl succinate	Fruity, melon	1200	<1	4.35 ± 0.36 b	5.99 ± 0.49 a	1.73 ± 0.20 e	ND	2.37 ± 0.19 d	3.17 ± 0.33 c
Ethyl 9-decenoate	Fruity	0.1	1700<	314.07 ± 26.91 b	250.42 ± 24.63 b	176.10 ± 16.83 c	255.14 ± 24.16 b	384.59 ± 36.53 a	266.05 ± 27.28 b
Isobutyl decanoate	Fruity			1.63 ± 0.13 c	0.76 ± 0.09 d	4.44 ± 0.38 a	1.84 ± 0.20 c	ND	3.39 ± 0.36 b
Methyl salicylate	Pepper, mint			90.13 ± 10.11 b	82.55 ± 8.30 b	122.72 ± 13.76 a	84.05 ± 7.43 b	105.93 ± 9.96 ab	119.99 ± 13.03 a
Ethyl 4-hydroxybutanoate				4.66 ± 0.39 c	10.70 ± 1.05 a	2.03 ± 0.19 d	8.00 ± 0.58 b	4.69 ± 0.49 c	1.92 ± 0.22 d
2-Phenylethyl acetate	Fruity, rose	1.8	64–264	281.44 ± 26.43 b	118.13 ± 12.67 c	289.61 ± 24.96 b	115.17 ± 13.55 c	241.95 ± 23.69 b	475.72 ± 44.85 a
Ethyl dodecanoate	Oily, fatty, fruity	1.5	37–87	96.54 ± 9.36 c	131.15 ± 12.13 b	105.19 ± 10.52 bc	204.28 ± 20.66 a	56.06 ± 5.73 d	116.89 ± 11.71 bc
Isoamyl decanoate				16.92 ± 1.58 b	8.26 ± 0.90 d	11.43 ± 1.36 c	30.41 ± 3.13 a	2.27 ± 0.28 e	12.55 ± 1.17 c
Ethyl hexadecanoate				43.28 ± 3.99 a	39.39 ± 3.68 a	16.85 ± 1.55 b	43.46 ± 3.83 a	22.61 ± 2.30 b	20.93 ± 2.08 b
Ethyl 9-hexadecenoate				3.11 ± 0.28 a	1.49 ± 0.17 b	1.67 ± 0.16 b	1.78 ±0.13 b	ND	1.57 ± 0.14 b
Ethyl octadecanoate				5.21 ± 0.39 a	3.69 ± 0.40 b	3.35 ± 0.38 b	4.56 ± 0.37 a	2.65 ± 0.29 c	2.56 ± 0.23 c
∑Ester				7912.38 ± 778.81 a	4218.86 ± 398.50 c	8758.95 ± 857.68 a	5693.95 ± 547.77 b	6232.68 ± 588.30 b	8178.61 ± 765.18 a
Monoterpene									
Terpinolene				ND	ND	ND	5.87 ± 0.60 b	12.25 ± 1.13 a	4.04 ± 0.39 c
Nerol	Violets, floral	0.5	3–15	ND	ND	ND	7.73 ± 0.69 a	ND	1.50 ± 0.08 b
p-Cymene	Lemon, fruity	11.4	1	ND	ND	ND	ND	ND	14.99 ± 1.36
Linalool	Flowery, Muscat	0.025	158–943	1.00 ± 0.08 e	3.96 ± 0.35 d	8.38 ± 0.76 b	23.57 ± 1.99 a	6.32 ± 0.58 c	6.70 ± 0.59 c
4-Terpineol				ND	ND	ND	ND	ND	29.50 ± 2.58
α-Terpineol	Lilac, floral, sweet	0.25	5–47	ND	1.19 ± 0.13	ND	6.19 ± 0.63 b	11.70 ± 1.26 a	4.68 ± 0.39 c
β-Citronellol	Rose	0.1	67–143	6.89 ± 0.67 d	ND	8.78 ± 0.79 c	10.37 ± 1.07 b	6.67 ± 0.62 d	14.33 ± 1.34 a
Geraniol	Citric, geranium	0.02	88–304	1.76 ± 0.20 d	3.05 ± 0.28 c	4.06 ± 0.43 b	3.04 ± 0.40 c	ND	6.09 ± 0.58 a
∑Monoterpene				9.64 ± 0.95 e	8.19 ± 0.76 e	21.22 ± 1.98 d	56.75 ± 5.38 b	36.94 ± 3.59 c	81.84 ± 7.31 a

Different letters within the same column indicate significant differences (*p* < 0.05). ND, not detected. Odor description and threshold values are derived from previous studies [32,63,64,65,66]. Odor activity value (OAV) was calculated by dividing the concentration of the compound by the odor threshold value of the compound. Only the lowest and highest values are shown in the table.

## Data Availability

The data used to support the findings of this study can be made available by the corresponding author upon request.

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
