# Peer review of "Quality Characteristics and Antioxidant Activities of Six Types of Korean White Wine"

_foods, 2023, doi:10.3390/foods12173246_

Round 1

Reviewer 1 Report

I think that the statements should include wines, as 6 Korean wines are being studied, not just one.

 Line 128, Evaluation of Antioxidant Activities of Korean White Wine, The mixture was stirred thoroughly and placed at room temperature in the dark for 10 min. The decrease in absorbance of the resulting solution was observed at 517 nm after 10 min. This reaction time seems to me to be too short, it should be at least 30 min or 60 min.

from line 203 to 207, it does not correspond to results, it would be more introduction and something already mentioned above, it is not necessary.

Line 309, It would be necessary to compare the results of the 6 wines made with other white wines, in order to confirm that these results are as expected and are not influenced by the short reaction time before measurement.

Line 342, volatile compounds, I miss in the discussion again the comparison of the results of the white wines studied with other wines made with the same strains.

Author Response

Reviewer 1

I think that the statements should include wines, as 6 Korean wines are being studied, not just one.

 Line 128, Evaluation of Antioxidant Activities of Korean White Wine, The mixture was stirred thoroughly and placed at room temperature in the dark for 10 min. The decrease in absorbance of the resulting solution was observed at 517 nm after 10 min. This reaction time seems to me to be too short, it should be at least 30 min or 60 min.

→ Thanks for pointing out it. We agree that the reaction time is too short so we reanalyzed DPPH of white wines and revised the values in the manuscript.

from line 203 to 207, it does not correspond to results, it would be more introduction and something already mentioned above, it is not necessary.

→ We moved these sentences into introduction and slightly revised them. Thanks.

Line 309, It would be necessary to compare the results of the 6 wines made with other white wines, in order to confirm that these results are as expected and are not influenced by the short reaction time before measurement.

→ As a result of reanalyzed DPPH values, we confirmed that the short reaction time led to incorrect results. We revised the DPPH result in the manuscript. Additionally, we added some references that compared the antioxidant activity of different grape varieties. Thanks.

Line 342, volatile compounds, I miss in the discussion again the comparison of the results of the white wines studied with other wines made with the same strains.

→ Thanks for a good comment. This study compares the quality characteristics of white wines made from different Korean grape varieties, using the same yeast strain (S. cerevisiae EC1118). As the yeast strain used was consistent across the study, specific details regarding the fermentation process were not provided. Notably, the yeast strain's properties do not constitute a significant aspect of this research. Moreover, since the aromatic attributes of the wines, contingent on grape variety, remain independent of the yeast strain, referencing studies from other countries utilizing the same yeast strain for distinct grape wines could potentially lead to reader confusion. Considering these factors, we believe that refraining from such references better aligns with the study's objectives. However, if you still find it necessary to include this aspect in the discussion, we will incorporate it in the subsequent revision. Thank you for your consideration.

Reviewer 2 Report

The article presents a physicochemical characterization of grape musts and wines made from six white grape cultivars. It is well structured and the methodology is well explained in all cases. However, there are several comments that the authors should address in order to improve the manuscript and make it suitable for publication.

From 3. Results and Discussion section on, please avoid the repetition of “Produced Using Six types of White Grape cultivar” in each and every section. The reader know the materials employed after section 2.

1. Introduction

Lines 28-36. There is a vague introduction to the general winemaking process, mentioning only whites and reds. It should be left out other major processes, which are becoming more and more important, such as the production of rosé wine, or even orange wine. In addition, several de facto statements are made along these lines without any bibliographical citation, which should be added.

Line 51. Reference is made to the climatology of the Korean region. It would be appropriate to mention the problems associated with high temperatures, humidity and rainfall in the cultivation of vines; especially the problems associated with the presence of pests and diseases associated with the cultivation of Vitis vinifera

Line 67: Green grape cultivars? Is it white?

Line 69. Avoid personal forms like “our”

2. Materials and methods

2.1. Strain, Material and culture Conditions

Important information regarding vine plot is missing in this section. Also, information about the climate in the region would be beneficial as it significantly influences the ripening process of the grapes and the physicochemical composition of their musts. What rootstock was used? Planting frame? Are irrigation treatments carried out? Are phytosanitary treatments applied? All this information should be mentioned in this section given the importance that the absence or presence of treatments and vegetation management can have on grape composition.

When were the grapes harvested? On the basis of what criteria? Were ripening controls carried out? Were all varieties harvested at the same time? Do they all have a similar phenological cycle or are there differences between them?

Additionally, the use of prime names is highly recommended in the scientific literature. These should be consulted in the database to provide the number of the variety under study. Also, Seibel is a Vitis interspecific crossing and there are currently more than 1800 varieties coded with this name, which one is used? As for the varieties Kerner and Vidal, there are 2 and 46 varieties catalogued with this name respectively, which ones were used? Finally, it would be advisable to use the name Chungsoo as it is the prime name and Cheongsoo is a synonym. Pleas, check Vitis International Variety Catalogue.

2.2. Wine fermentation

Line 82: Why were grapes washed? Not a common practice in European regions

Line 83: The dose of Sulphur dioxide is closely linked to the state of health of the grapes at the time of harvest. What was the state of health of the grapes?

Line 88: Why units of measurement such as kilograms are used when referring to a liquid medium. The use of liters would be preferable.

How were the grapes pressed? Since the type of press and the pressure applied greatly influences the composition of the grapes, this treatment should be indicated.

2.3. Total Flavonoid Contents in Korean White Wine

Line 126: How much time was it vortexed?

2.8. Sensory evaluation

How many days after bottling was it performed? Was the sensory panel previously trained? Please, specify.

3. Results and Discussion

3.1. Physicochemical Characteristics of Grape Must and Wine Produced Using Six types of White Grape cultivar

Lines 203-210. Looks like an introduction section. This is neither a result nor a discussion.

Table 1

Why are the total acidity results expressed as a percentage? It is much more practical for the reader if they are expressed in grams per liter, either in units of tartaric or sulphuric acid.

Do the authors have values for gluconic acid? This organic acid is a good indicator of the state of health of the grapes at the time of harvesting. If they have the data or samples to quantify it, it would be advisable for it to appear in table 1.

Line 254: The term ripeness/ripening is preferable to maturity.

Table 2

The alcohol content for SB and CS is shown to be unusually low (about 10.5). Given the absence of residual sugars in fermentation, were these grapes harvested unusually early before full ripening? If so, this early harvest could affect the polyphenolic and aromatic content of these varieties.

Additionally, wines with 10.5% alcohol present problems in their microbiological stabilization, as well as unsatisfactory results in their sensory analysis, at least in the USA and Europe.

3.3. Antioxidant Activity if Korean White Wine Produced using Six Types of White Grape Cultivars

Line 317-319: It is mentioned that Sulphur affects the measures of antioxidant power. However, the results do not take into account the applied dose of 200 mg/5 kg of grapes. Again, this dose is highly variable depending on the pressure applied in the grape pressing and the yield of the process. Data in mg/L (international system units) would be preferable. On the other hand, the antioxidant capacity of red grape varieties is mentioned on several occasions; it would be advisable to provide some specific value and its bibliographical citation in order to facilitate the reader's understanding.

3.5. Volatile Aromatic Compounds in Korean White Wine Produced using Six Types of White Grape Cultivars

Lines 393-394: Since the odor threshold is indicated, it would be beneficial if the indication of the Odor Activity Values would appear in the article. Try to calculate the OAV as the ratio between the concentration of each compound and its perception threshold. See Amores-Arrocha et al., (2018). Food Res. Intl 105, 197-209.

Figure 4

Providing the values of the loadings for the Principal Component would be beneficial to the understanding of the figure as it would allow the reader to know which compound or compounds have the most positive or negative influence on each Principal Component.

3.6. Sensory Evaluation of Korean White Wine Produced Using Six Types of White Grape Cultivars

Figure 5

Displaying the sensory analysis results in a spider web diagram would be beneficial, since only significant differences between cultivars would be appreciated by a glance.

Only a minor spell check and English editing is required. Some sentences should be shortened in order to be easier to understand by the readers. 

Author Response

Thanks for your valuable comments,

I have attached my response letter in the accompanying file for your review. I would greatly appreciate it if you could take a look. Thank you.

Reviewer 3 Report

In line 73: Please, provide information of the grapes health status and how they were determined? Also, data on the yield and cultivation form of the vines are needed!  

In line 233: Please, explain the increase in total acidity. Is it under the influence of yeast?  

In line 243: Please explain the increase in tartaric acid.  

In Table 2: Explain the presence of sucrose in the wine. Sucrose was not previously detected in the grape must!  

In line 375: These compounds (isoamyl alcohol and 2-phenilethanol) were also detected in other wines made from autochthonous grape varieties. I highly recommend citing the reference Journal of the Serbian Chemical Society 2023 Volume 88, Issue 1, Pages: 11-23
https://doi.org/10.2298/JSC220311056M

Author Response

I have provided answers to your questions below for your reference. I would appreciate it if you could review them. Thank you.

In line 73: Please, provide information of the grapes health status and how they were determined? Also, data on the yield and cultivation form of the vines are needed!  

→ We harvested only 10 kg of each grape and thoroughly selected the healthy grapes. Gluconic acid was not detected, ensuring the grapes were healthy. Analyzing the yield and cultivation form of the grapes are not our field. We also agree that further research on those of white grapes grown in Korea is needed and some other group will deal with that part in the future. Thanks.

In line 233: Please, explain the increase in total acidity. Is it under the influence of yeast?  

→ Yes, increased total acidity is influenced by metabolism of yeast. We revised the sentence. Thanks.

In line 243: Please explain the increase in tartaric acid.  

→ We miscalculated the peak area of tartaric acid in grape samples. We checked the HPLC data again, recalculated them, and revised the tartaric acid content in the manuscript. Thanks. 

In Table 2: Explain the presence of sucrose in the wine. Sucrose was not previously detected in the grape must!  

→ Sorry, it was our mistake. We misread the retention time of the HPLC data. Sucrose was not detected in all the wine. We removed sucrose in Table 1 and 2. Thanks.

In line 375: These compounds (isoamyl alcohol and 2-phenilethanol) were also detected in other wines made from autochthonous grape varieties. I highly recommend citing the reference Journal of the Serbian Chemical Society 2023 Volume 88, Issue 1, Pages: 11-23
https://doi.org/10.2298/JSC220311056M

→ We cited that reference. Thanks.

Reviewer 4 Report

This is an interesting report about the use of six grape varieties for production of Korean wine. The use of grapes at Korea still unusual compared with other fruits. There is a lack of information about the potential of grapes varieties for the production of Korean wines.  

The introduction covers some basic concepts about wine production. It should try to talk more about the composition of wines, namely their phenolic compounds and aroma compounds. According to the characterization that was carried out to the wines in this study.

The presented results are only described, not having the same discussion. Although there are few data available on the characterization of Korean wines, the authors should compare the results obtained with others existing in the bibliography. For example, compare the results obtained with wines of the same varieties produced in other countries.

Author Response

Dear Reviewer 4,

We greatly appreciate your thorough review to enhance the completeness of my paper. We have carefully incorporated your comments to improve the revised manuscript. Kindly review the attached response at your convenience. 

Sincerely

Round 2

Reviewer 2 Report

Dear Author. 

After the first round of revisions, the manuscript has been substantially improved following the comments made. 

However, there are some minor errors that should be corrected. 

Line 75: You continue to use the term "we", which is a personal form. It is advisable to use passive sentences or reflexive verb forms. Also, I do not understand the use of the form "we" when there is only one author. 

Section 2.1: There is still a lack of interesting agronomic information, which I think is easy to obtain even if the vineyard is privately owned (What rootstock was used? Planting frame? Are irrigation treatments carried out? Are phytosanitary treatments applied?)

Section 3.3: The reviewer has some doubts about the author's statement that the presence of SO2 does not affect the antioxidant capacity result, since there is always sulphur remaining, both in free and combined state, at the end of fermentation.

Author Response

Thanks for your valuable comments.

I have revised the manuscript according to the comments and highlighted them in blue. Please check the response below and review the revised manuscript once again.

Sincerely, 

---------------------------------------------------------------------------------

Line 75: You continue to use the term "we", which is a personal form. It is advisable to use passive sentences or reflexive verb forms. Also, I do not understand the use of the form "we" when there is only one author. 

 → I used the term ‘we’ habitually. Now I have revised the manuscript to eliminate the use of the term ‘we’ throughout.

Section 2.1: There is still a lack of interesting agronomic information, which I think is easy to obtain even if the vineyard is privately owned (What rootstock was used? Planting frame? Are irrigation treatments carried out? Are phytosanitary treatments applied?)

  → The agronomic information was added to the manuscript. Thanks.

Section 3.3: The reviewer has some doubts about the author's statement that the presence of SO2 does not affect the antioxidant capacity result, since there is always sulphur remaining, both in free and combined state, at the end of fermentation.

 → According to the Morgan et al. (2019), the concentration of free SO2 decreases over the fermentation. Since only free form of SO2 can have antioxidant activity, I think SO2 treatment doesn’t affect antioxidant activity of final white wine. Additionally, if there is any free SO2 is remained at the end of fermentation, I think current result (VB wine has the highest antioxidant activity) is reliable because same amount of SO2 was treated and fermentation conditions were same. Thanks.

Morgan et al. (2019) Response to sulfur dioxide addition by two commercial Saccharomyces cerevisiae strains. Fermentation, 5, 69.

Reviewer 3 Report

I am satisfied with the accepted changes. Thank you.

Author Response

Thank you so much.